# IT³: Idempotent Test-Time Training

**Nikita Durasov** [* 1]  **Assaf Shocher** [* 2]  **Doruk Oner** [3]  **Gal Chechik** [2]  **Alexei A. Efros** [4]  **Pascal Fua** [1]

## Abstract

Deep learning models often struggle when deployed in real-world settings due to distribution shifts between training and test data. While existing approaches like domain adaptation and test-time training (TTT) offer partial solutions, they typically require additional data or domain-specific auxiliary tasks. We present Idempotent Test-Time Training (IT³), a novel approach that enables on-the-fly adaptation to distribution shifts using only the current test instance, without any auxiliary task design. Our key insight is that enforcing idempotence—where repeated applications of a function yield the same result—can effectively replace domain-specific auxiliary tasks used in previous TTT methods. We theoretically connect idempotence to prediction confidence and demonstrate that minimizing the distance between successive applications of our model during inference leads to improved out-of-distribution performance. Extensive experiments across diverse domains (including image classification, aerodynamics prediction, and aerial segmentation) and architectures (MLPs, CNNs, GNNs) show that IT³ consistently outperforms existing approaches while being simpler and more widely applicable. Our results suggest that idempotence provides a universal principle for test-time adaptation that generalizes across domains and architectures.

poster / code / video / web

## 1. Introduction

Supervised learning methods, while powerful, typically assume that training and test data come from the same distribution. Unfortunately, this is rarely true in practice. Data

This work was supported in part by the Swiss National Science Foundation. *Equal contribution [1]CVLAB, EPFL [2]NVIDIA [3]NeuraVision Lab, Bilkent University [4]UC Berkeley. Correspondence to: Nikita Durasov <nikita.durasov@nvidia.com>.

*Proceedings of the 42$^{nd}$ International Conference on Machine Learning*, Vancouver, Canada. PMLR 267, 2025. Copyright 2025 by the author(s).

encountered by systems operating in the real world often differs substantially from what they were trained on due to data distribution shifts over time or other changes in the environment. This inevitably degrades performance, even in state-of-the-art models (Recht et al., 2018; Hendrycks et al., 2021; Yao et al., 2022). Machine learning systems used in production not only need to adapt to distribution shifts but also must do so on-the-fly using very limited data.

Thus, in this work, we focus on adapting to distribution shifts on-the-fly using only the current test instance or batch, without access to any additional labeled or unlabeled data during inference. During training, the model only has access to the base distribution training data, without any knowledge of the test distribution, which may be different.

Adversarial robustness and domain adaptation address related challenges. However, they typically require additional data either during training or inference, and sometimes rely on specific assumptions about the nature of the shift. While effective when such additional knowledge is available, they are not designed for immediate, instance-level adaptation and are not applicable in the scenario we envision. Test-Time Training (TTT) (Sun et al., 2020) offers a promising alternative by adapting the model during inference by performing an auxiliary self-supervised task on each test sample. This enables the model to handle corrupted and Out-of-Distribution (OOD) data using only the current test instance or batch, without access to anything else. However, TTT requires performing an auxiliary task specific to the data modality, such as orientation prediction or inpainting for image data (Gandelsman et al., 2022). And defining an appropriate auxiliary task is not straightforward in general.

In this paper, we argue that enforcing *idempotence* can profitably replace the auxiliary tasks in TTT and yields an approach we dub IT³ that is a versatile and powerful while generalizing well across domains and architectures.

More specifically, let $f$ be a deep network that takes as input a vector $\mathbf{x}$ and a second auxiliary variable that can either be the ground truth label $\mathbf{y}$ corresponding to $\mathbf{x}$ or a neutral uninformative signal $\mathbf{0}$. In (Durasov et al., 2024b), it was shown that if such a network is trained so that $f(\mathbf{x}, \mathbf{0}) = f(\mathbf{x}, \mathbf{y}) = \mathbf{y}$, then at test time the distance $||f(\mathbf{x}, f(\mathbf{x}, \mathbf{0})) - f(\mathbf{x}, \mathbf{0})||$ correlates strongly with the prediction error. What if, at test time, we could actively minimize this distance

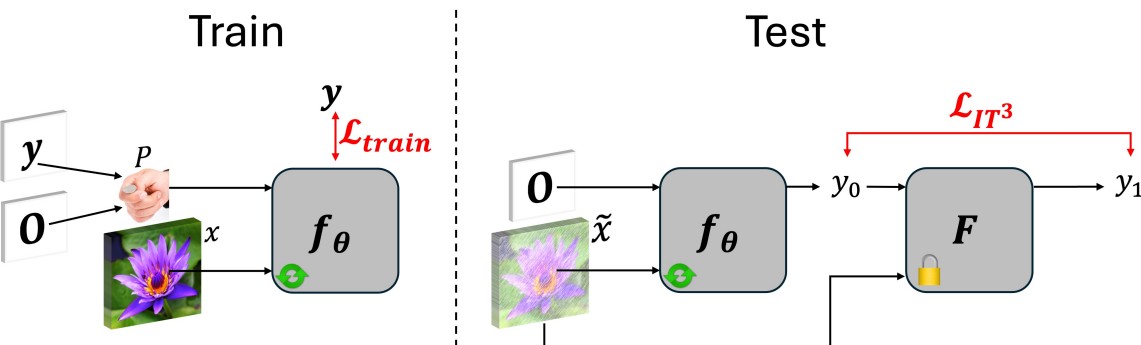

*Figure 1.* **Idempotent Test-Time Training (IT³) approach.** During training (left), the model $f_\theta$ is trained to predict the label $y$ with or without $y$ given to it as input. At test time (right), when given a corrupted input, the model is sequentially applied. It then briefly trains with the objective of making $f_\theta(\mathbf{x}, \cdot)$ to be idempotent using only the current test input.

whenever we encounter an OOD instance? Could we "pull it" into the distribution? IT³ uses this distance as a loss for TTT sessions. Unfolding $y_0, y_1$ we obtain: $||f(\mathbf{x}, f(\mathbf{x}, \mathbf{0})) - f(\mathbf{x}, \mathbf{0})||$. This makes $f(\mathbf{x}, \cdot)$ *idempotent*. Fortunately, it has been shown that, while not trivial, training a model to achieve idempotence is feasible (Shocher et al., 2024). See further discussion in Appendix B.

This yields a generic method that does not rely on any specific domain properties. This is in contrast to prior TTT methods that rely on a domain specific auxiliary task. By leveraging the universal property of idempotence, IT³ can adapt OOD test inputs on-the-fly across various domains, tasks and architectures. This includes image classification with corruptions, aerodynamic predictions for airfoils and cars, tabular data with missing information, age prediction from faces, and large-scale aerial photo segmentation, using MLPs, CNNs, or GNNs.

## 2. Related Work

In essence, IT³ relies on *idempotence* to generalize *Test-Time Training*. We briefly review these two fields.

### 2.1. Test-Time Training (TTT)

The idea of leveraging test data for model adaptation dates back to methods like transductive learning (Gammerman et al., 1998). Early approaches, such as transductive SVMs (Collobert et al., 2006) and local learning (Bottou & Vapnik, 1992), aimed to adapt predictions for specific test samples rather than generalizing across unseen data.

Training neural networks solely on single test instances, without pre-training, has been demonstrated in the "deep internal learning" line of work, for many image enhancement tasks (Shocher et al., 2018; Gandelsman et al., 2019) and single image generative models (Shocher et al., 2019; Shaham et al., 2019).

**Distribution Shifts**: TTT has emerged as a solution to the problem of generalization under distribution shifts. Using a pre-trained network and at test-time refining on a single instance each time. In the foundational work of Sun et al. (2020), the model is adjusted in real-time by solving an auxiliary self-supervised task, such as predicting image rotations, on each test sample. This on-the-fly adaptation has proven good at improving robustness on corrupted and Out-Of-Distribution (OOD) data. As the self-supervised learning methods became more efficient (He et al., 2022), they could be exploited for TTT (Gandelsman et al., 2022). Extensions such as TTT++ (Liu et al., 2021) require access to the entire test set. TENT (Wang et al., 2021) adapts during inference at the batch level, based on batch entropy, but cannot be applied to single instances or very small batches. Moreover, it relies on updating the model's normalization layers, making it architecture dependent. Another problem that many existing TTT approaches face is their task dependency, meaning they are designed to work for a specific data types, which is almost always image classification. The recent method ActMAD (Mirza et al., 2023) addresses this limitation by pulling the mean and variance of data embeddings closer to those of the training data, improving predictions on out-of-distribution or corrupted data. While its effectiveness has been mostly demonstrated on image data, the approach has the potential to be applied to other types of data as well and we will use as one of the baselines we compare against. In (Park et al., 2024), the authors propose a test-time adaptation method for depth completion, fine-tuning a specific adaptor layer using a consistency loss between two predictions. However, the approach is tailored to depth completion and not applicable to other tasks.

**TTT vs. Test Time Adaptation (TTA)**: TTT operates per-instance, with no assumption that future test data will be similar. In contrast, TTA (Liang et al., 2024) adapts using a *large* test set from the same shift (Batch/Domain TTA), or assuming correlation between instances (Online TTA).

Most previous work (Sun et al., 2020; Gandelsman et al., 2022) have thus treated TTA and TTT as distinct paradigms rather than direct competitors. TTA exploits abundant test data but cannot tune the training process, while TTT shapes training but handles every test instance in isolation. Each approach suits different scenarios.

## 2.2. Idempotence in Deep Learning

Idempotence, a concept rooted in mathematics and functional programming, refers to an operation whose repeated application yields the same result as a single application. Mathematically, for a function $f$, being idempotent means

$$f(f(x)) = f(x), \quad \forall x . \tag{1}$$

In other words, applying the function multiple times has no effect beyond the first one. In the context of linear operators, idempotence corresponds to orthogonal projection. Over $\mathbb{R}^n$, these are matrices $A$ that satisfy $A^2 = A$, with eigenvalues of either 0 or 1; they can be interpreted as geometrically preserving certain components while nullifying others. This concept was recently applied in generative modeling. Idempotent Generative Network (IGN) (Shocher et al., 2024) is a generative model that maps data instances to themselves, $f(x) = x$, and maps latents to targets that also map to themselves, $f(f(z)) = f(z)$. It was shown to 'project' corrupted images onto the data manifold, effectively removing the corruptions without prior knowledge of the degradation.

Energy-Based Models (EBMs; (Ackley et al., 1985)) offer a related perspective by defining a function $f$ that assigns energy scores to inputs, with higher energy indicating less desirable or likely examples, and lower energy indicating those that fit the model well. IGN introduces a similar concept but frames it differently: Instead of $f$ directly serving as the energy function, the energy is implicitly defined via the difference $\delta(y) = D(f(y), y)$, where $D$ measures the distance between the model's prediction and its input. In this framework, training $f$ to be idempotent minimizes $\delta(f(z))$, pushing the model toward a low-energy configuration where its outputs remain stable under repeated applications. Thus, $f$ can be interpreted as a transition operator that drives high-energy inputs toward a low-energy, stable domain, reducing the need for separate optimization procedures to find the energy minimum.

In concurrent work, the ZigZag method has first been proposed and then extended to recursive networks (Durasov et al., 2024a;b). It introduces idempotence as a means to assess uncertainty in neural network predictions. ZigZag operates by recursively feeding the model's predictions back as inputs, allowing the model to refine its outputs. The consistency between successive predictions acts as an uncertainty metric, where stable, unchanged outputs indicate higher confidence, while divergent predictions signal uncertainty or out-of-distribution (OOD) data. Unlike popular sampling-based uncertainty estimation methods (Gal & Ghahramani, 2016; Lakshminarayanan et al., 2017; Wen et al., 2020; Durasov et al., 2021), ZigZag does not require many forward passes or complex sampling, making it more computationally efficient for real-time applications.

## 3. Method

Given a pre-trained model, IT³ aims to dynamically adapt its weights at inference time using Test-Time Training (TTT) to reduce uncertainty and handle Out-of-Distribution (OOD) instances. As discussed above, other TTT approaches rely on satisfying domain specific auxiliary tasks to achieve this. Instead, we rely on the model we train being idempotent on the training set and adapt its weights at inference time to approach idempotence on the test set as new samples are being fed to it. This pulls the representations of OOD inputs back into the distribution of the training data and improves the model's performance on corrupted or OOD instances.

In this section, we describe how we make our models idempotent for training set samples, how we use the idempotence loss for TTT during inference, and how we adapt the algorithm for online scenarios.

### 3.1. Making the Network Idempotent at Training Time

Let $f_\theta$ be a generic network with weights $\theta$ that takes an input $\mathbf{x}$. We wish to deploy in an environment where the statistical distribution of the samples it receives may change over time. To this end, as in ZigZag (Durasov et al., 2024a), we modify slightly its input layer so that it can accept a second argument that be either $\mathbf{y}$, the desired output of the network given input $\mathbf{x}$, or a neutral uninformative signal $\mathbf{0}$. During the initial training, we minimize the supervised loss

$$\mathcal{L}_{\text{train}} = \|f_\theta(\mathbf{x}, \mathbf{y}) - \mathbf{y}\| + \|f_\theta(\mathbf{x}, \mathbf{0}) - \mathbf{y}\| , \tag{2}$$

as depicted on the left side of Fig. 1. This enforces

$$\mathbf{y}_0 = f_\theta(\mathbf{x}, \mathbf{0}) \approx \mathbf{y} , \tag{3}$$

$$\mathbf{y}_1 = f_\theta(\mathbf{x}, \mathbf{y}_0) \approx f_\theta(\mathbf{x}, \mathbf{y}) \approx \mathbf{y} , \tag{4}$$

$$\Rightarrow f_\theta(\mathbf{x}, f_\theta(\mathbf{x}, \mathbf{0})) \approx f_\theta(\mathbf{x}, \mathbf{0}) .$$

In other words, $\theta$ has been adjusted so that function $f_\theta(\mathbf{x}, \cdot)$ is as idempotent as possible for all $\mathbf{x}$ in the training set. Of course, when $\mathbf{x}$ is comes from the test set, there is no guarantee of that because there may be distribution shift between the two sets. In (Durasov et al., 2024a), it is shown that the deviation from the equality of Eq. 4 expressed as $\|f_\theta(\mathbf{x}, f_\theta(\mathbf{x}, \mathbf{0})) - f_\theta(\mathbf{x}, \mathbf{0})\|$ correlates strongly with the accuracy of the prediction and can be used to detect testing samples that are out-of-distribution with respect to the training set. Fig. 2 illustrates this in the case of a network trained

to predict the lift-over-drag ratio (L/D) of a 2D airfoil. In other words, here, $\mathbf{x}$ is a 2D outline representing an airfoil and the output $y$ is expected the corresponding L/D.

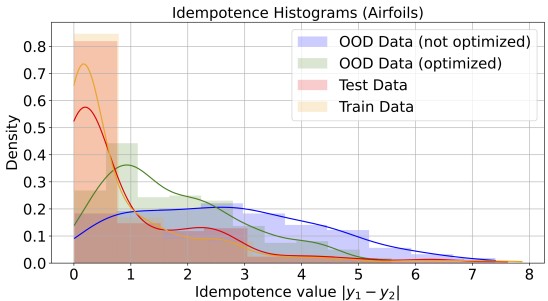

*Figure 2.* **Idempotence vs. Out-of-Distributionness**: We plot the distribution of idempotence errors, measured by the distance $|y_1 - y_2|$ in Eq.4, for training, test, and OOD data. For OOD samples, we show the errors both before and after minimizing them globally. OOD samples exhibit significantly larger idempotence errors, which decrease after optimization. Figuratively, IT³ pushes the OOD representations to be more similar to those of the training distribution. In Sec.4, we show that this reduction yields improved performance.

### 3.2. Test-Time Training

In ZigZag (Durasov et al., 2024a), deviations from idempotence, as measured by the distance between the two predictions of Eq. 4, are used to evaluate the accuracy of a prediction. In IT³ , we propose to go further and to minimize these deviations at *inference* time to compensate for potential domain shifts between training and test data.

A naive way would be to minimize the loss function

$$\mathcal{L}_{\text{TTT}} = \|f_\theta(\mathbf{x}, f_\theta(\mathbf{x}, \mathbf{0})) - f_\theta(\mathbf{x}, \mathbf{0})\|, \quad (5)$$

for all samples $\mathbf{x}$ received as inference time. However, this can produce undesirable side effects. For instance, if $\mathbf{y}_0 = f_\theta(\mathbf{x}, \mathbf{0})$ is an incorrect prediction, minimizing $\|\mathbf{y}_0 - \mathbf{y}_1\|$ may cause $\mathbf{y}_1 = f_\theta(\mathbf{x}, \mathbf{y}_0)$ to be pulled toward the incorrect $\mathbf{y}_0$, thereby magnifying the error. Another potential failure mode is to encourage $f_\theta(\mathbf{x}, \cdot)$ to become the identity function, with is trivially idempotent.

To prevent this, we modify the test-time training procedure as shown on the right side of Fig, 1. We keep a copy of the model as it was at the end of the training phase, denoted as $F = f_\Theta$, where $\Theta$ are the weights obtained after the initial training of Sec. 3.1, which will not be updated further. We then take the test-time loss to be

$$\mathcal{L}_{\text{IT³}} = \|F(\mathbf{x}, f_\theta(\mathbf{x}, \mathbf{0})) - f_\theta(\mathbf{x}, \mathbf{0})\|, \quad (6)$$

where $f_\theta$ is the model being updated at test-time. Here, the first prediction $\mathbf{y}_0 = f_\theta(\mathbf{x}, \mathbf{0})$ is computed as before, but the second one, $\mathbf{y}_1 = F(\mathbf{x}, f_\theta(\mathbf{x}, \mathbf{0}))$, is made using the frozen

model $F$. By updating only $f_\theta$ and keeping $F$ fixed, we ensure that $\mathbf{y}_0$ is adjusted to minimize the discrepancy with $\mathbf{y}_1$, without pulling $\mathbf{y}_1$ toward an incorrect $\mathbf{y}_0$. A similar idea was employed in the IGN approach (Shocher et al., 2024) when meaningful predictions are required. After each TTT optimization iteration, the dynamic model $f_\theta$ is initialized with $\Theta$, ready for the next input.

Essentially, IT³ extends the projection principle of Idempotent Generative Networks (IGN) (Shocher et al., 2024). IGNs map corrupted inputs onto the distribution of valid data by enforcing idempotence. Similarly, IT³ projects OOD $(x, y)$ pairs onto the distribution of valid ones by iteratively refining the network's internal representations. While only $y$ explicitly changes, every layer's activations—functions of both $x$ and $y$—adjust to better fit the distribution of in-distribution representations, much like IGN corrects corrupted data by pulling it toward the natural image manifold. See detailed discussion in Appendix B.

### 3.3. Online IT³

We introduce a variant of IT³ for a *different scenario*: Given data streams, where the distribution shifts continuously over time, in a continual learning setup, we modify IT³ to operate in an online mode by not resetting $f_\theta$ back to $F$ after each TTT episode, as we did in Section 3.2. We essentially assume that the distribution mostly shifts smoothly and, thus, there is a good reason to believe that the current state of $f_\theta$ is a better initialization for the next TTT episode than the original $F$. This makes the model evolve over time. In this scheme, it can happen that the performance of the model on data from its original training decreases significantly, a phenomenon known as catastrophic forgetting (Kirkpatrick et al., 2017). This is acceptable as the goal is to perform well on data at the present moment, rather than on past examples.

We make another modification in the second sequential application of the model, that is, when computing $F(\mathbf{x}, f_\theta(\mathbf{x}, \mathbf{0}))$. Since the data keeps shifting, there is no reason to retain the frozen $F$ as an anchor indefinitely. Over time, $f_\theta$ may diverge far from $F$, making it irrelevant. Relying on the old state of the model would prevent the model from evolving efficiently. Replacing it with the current state of $f_\theta$ is out of the question, as it could causes collapse as described in Section 3.2. Instead, we need an anchor that is influenced by a reasonable amount of data, yet evolves over time. Our solution is to replace $F$ with an Exponential Moving Average (EMA) of the model $f_\theta$, denoted as $f_{\text{EMA}}$. This means $f_{\text{EMA}}$ is a smoothed version of $f_\theta$ over time. The test-time loss in the online setting then becomes

$$\mathcal{L}_{\text{online}} = \|f_{\text{EMA}}(\mathbf{x}, f_\theta(\mathbf{x}, \mathbf{0})) - f_\theta(\mathbf{x}, \mathbf{0})\|. \quad (7)$$

By updating both $f_\theta$ and $f_{\text{EMA}}$ incrementally, with $f_{\text{EMA}}$ serving as a stable reference that changes more slowly, the

model adapts to gradual shifts without overfitting to noise or temporary anomalies.

# 4. Experiments

We evaluate our approach across a diverse set of data types and tasks, including age prediction, image classification, and road segmentation in the visual domain, as well as aerodynamics prediction using 3D data and tabular data experiments. In all these scenarios, we first train the model using the supervised approach of Section 3.1 and then perform the test-time training of Section 3.2. For each task, we design an OOD test set for evaluation, that is, data drawn from a shifted distribution with respect to that of the training set. The OOD data is divided into several levels, with higher levels representing data that is progressively further from the training distribution. We evaluate our method for each level, presenting the results as bar plots for different batch sizes. After running the algorithm on a particular batch, we reset the model to its original, non-updated weights before evaluating the next batch.

In our experiments, we compare our method against a non-optimized model to demonstrate the effectiveness of TTT approaches relative to a vanilla model, as well as other popular baselines. For the image classification task, we include the original TTT method and a newer more versatile approach, ActMAD (Mirza et al., 2023), which we described in Section 2 and apply across all other setups. To further assess the effectiveness of our approach, we also evaluate all baselines, except the vanilla model, using different batch sizes. Across all scenarios, our method degrades more slowly than the baselines as the domain shift between training and testing data increases. Additionally, in Appendix A, we provide inference time comparisons for the considered approaches.

## 4.1. Tabular Data

Tabular data consists of numerical features and corresponding continuous target values for regression tasks from the UCI tabular datasets (Bay et al., 2000). They are widely used in machine learning research to benchmark regression models. In our case, we use The Boston Housing dataset describes housing prices in the suburbs of Boston, Massachusetts. It includes various features related to socioeconomic and geographical factors that influence housing prices. We take a test set and gradually apply random feature zeroing with increasing probabilities of 5%, 10%, 15%, and 20% (4 mentioned levels of OOD). This random feature dropping simulates out-of-distribution (OOD) data by progressively altering the input features, making the data less similar to the original training distribution. As the probability of feature dropping increases, the data becomes more OOD, which lowers the model's accuracy. The trained

model is a simple Multi-Layer Perceptron (MLP) optimized using the Adam optimizer, and we observe that IT³ consistently degrades less compared to other baselines across all OOD levels as depicted in Fig. 3.

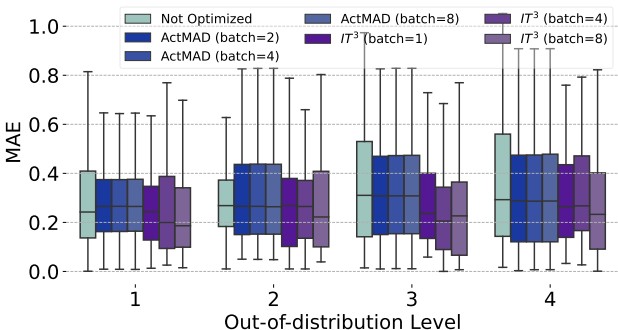

*Figure 3.* **UCI Results on OOD inputs**: The plots illustrate the performance of IT³ compared to other baselines across different OOD levels. The box plot for tabular data shows the distribution of MAE at various OOD levels, where $IT^3$ with different batch sizes ([**batch=1**, **batch=4**, **batch=8**]) degrades less compared to the Not optimized baseline and **ActMAD**. Larger batch sizes preserve performance more effectively.

## 4.2. CIFAR

We conducted similar experiments using the CIFAR-10 (Krizhevsky et al., 2014) dataset, selecting CIFAR-C (Hendrycks & Dietterich, 2019) as the out-of-distribution (OOD) data. CIFAR-C contains the same images as CIFAR-10 but with various common corruptions, such as Gaussian noise, blur, and contrast variations, simulating real-world conditions. These corruptions are applied at different severity levels, allowing us to evaluate how the model's performance degrades as the data shifts further from the original CIFAR-10 distribution. For this experiment, we used the *Deep Layer Aggregation* (DLA) (Yu et al., 2018) network, known for its strong performance in image classification and robustness to overfitting. We trained the model according to the guidelines from the original DLA paper to ensure optimal results. Fig.3 shows the evaluation error on CIFAR-C at severity level 5 for different types of corruptions, following (Sun et al., 2020). As shown, IT³ outperforms other baselines, with higher batch sizes yielding the best results.

## 4.3. Age Prediction

To experiment with image-based age prediction from face images, we use the UTKFace dataset (Zhang et al., 2017), a large-scale collection containing tens of thousands of face images annotated with age information. The model is trained on face images of individuals aged between 20 and 60, while individuals younger or older than this range are considered out-of-distribution (OOD) (Fig.5). The further the age is from the 20-60 interval, the higher the OOD level

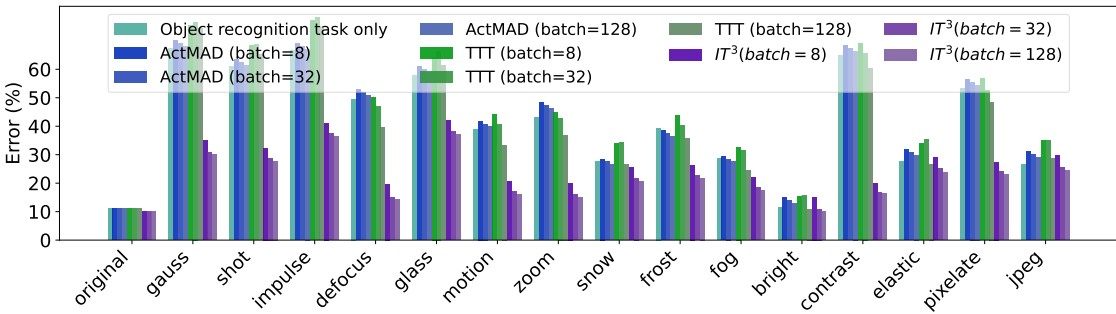

*Figure 4.* **Test error (%) on CIFAR-10-C with level 5 corruptions.** We compare our approaches, $IT^3$, with object recognition without self-supervision, TTT, and ActMAD. $IT^3$ improves over other baselines and higher batch size improves even further.

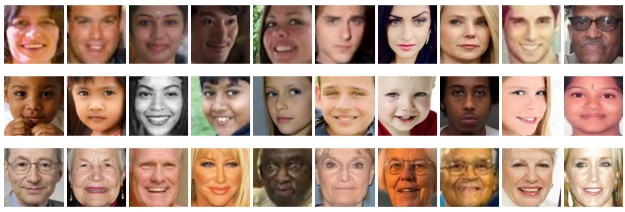

*Figure 5.* **Face Samples.** The **(top)** row shows training images of middle-aged individuals, while **(middle)** and **(bottom)** display images of older and younger individuals (OOD).

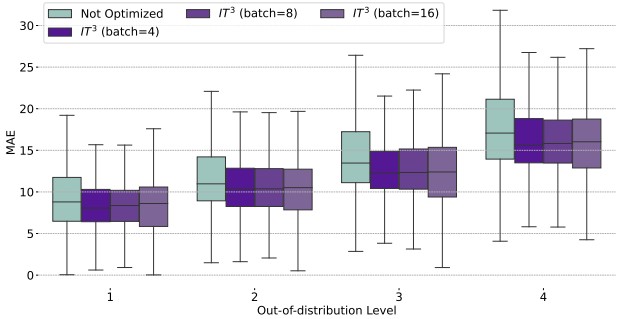

*Figure 6.* **Age boxplot results on OOD shapes.** Not optimized corresponds to a single model without TTT applied. $IT^3$ with [**batch=4**, **batch=8**, and **batch=16**] represents our method at different batch sizes. As the data shifts further from the training distribution, our method degrades less, with larger batches preserving performance more effectively.

we assign to it. We use a ResNet-152 backbone with five additional linear layers and ReLU activations. This architecture delivers strong accuracy, outperforming the popular ordinal regression model CORAL (Cao et al., 2020) and matching other state-of-the-art methods (Berg et al., 2021). We train our model on the UTKFace training set (limited to individuals aged 20-60) and then run inference on faces at different OOD levels. Again, $IT^3$ significantly outperforms the non-optimized model, as shown in Fig. 6.

### 4.4. Road Segmentation

Our method can be generalized to segmentation tasks as well. To demonstrate this, we consider the problem of

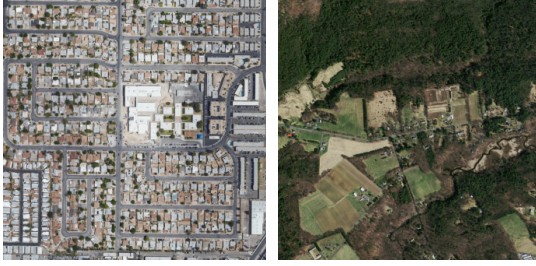

*Figure 7.* **Road Samples.** The roadTracer dataset **(left)** covers urban areas of six different countries while the Massachusetts dataset **(right)** primarily features rural neighborhoods along with some urban areas.

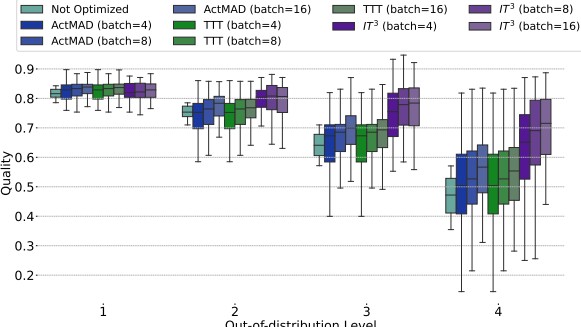

*Figure 8.* **Roads results on OOD images.** Not optimized corresponds to a single model without TTT applied. $IT^3$ with [**batch=4**, **batch=8**, and **batch=16**] represents our method at different batch sizes. As the data shifts further from the training distribution, our method degrades less compared to the Not optimized, TTT, and ActMAD, with larger batches preserving performance more effectively.

road segmentation in aerial imagery using the RoadTracer dataset (Bastani et al., 2018). We train a DRU-Net (Wang et al., 2019), on the RoadTracer dataset.

We perform OOD experiments using Massachusetts Road dataset (Mnih, 2013) that primarily comprises rural neighborhoods, as depicted in Fig. 7. We sample 450 images, each with dimensions of $1500 \times 1500$ pixels and divide them into four groups based on the Mean Squared Error (MSE) of the segmentation outputs, effectively creating dif-

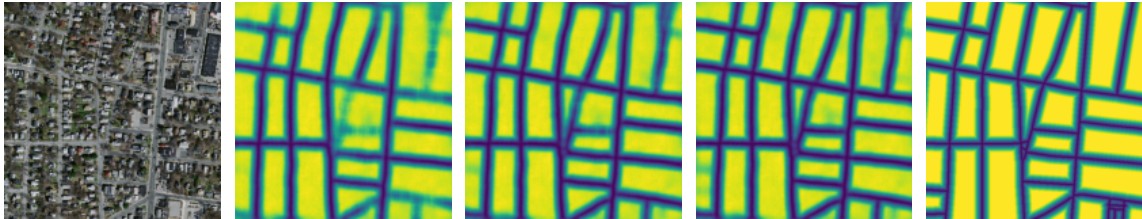

*Figure 9.* **Qualitative effect of IT³ on Road Segmentation.** From left to right: (1) Original aerial image, (2) Output before optimization, (3) IT³ output at the 5th iteration, (4) IT³ output at the 15th iteration, and (5) Ground truth label. The segmentation quality improves significantly with IT³ iterations, as observed in the progressively refined outputs at the 5th and 15th iterations.

ferent levels of distributional shift within the sampled set. We then further train the network on these OOD subsets using the ZigZag method (Durasov et al., 2024a).

We evaluate road segmentation performance by using *Correctness*, *Completeness* and *Quality* (CCQ) metric (Wiedemann et al., 1998) which is a popular metric to evaluate delineation performance. The *Correctness*, *Completeness* and *Quality* are equivalent to precision, recall and intersection-over-union, where the definition of a true positive has been relaxed from spatial coincidence of prediction and annotation to co-occurrence within a distance of 5 pixels. As shown in Fig.8, IT³ significantly improves performance on OOD images (see Fig.16 for more qualitative examples).

### 4.5. Aerodynamics Prediction

**Wings.** Our method is versatile and can handle various types of data. To illustrate this, we generated a dataset of 2,000 wing profiles, as depicted in Fig.10, by sampling the widely used NACA parameters (Jacobs & Sherman, 1937). We used the XFoil simulator (Drela, 1989) to compute the pressure distribution along each profile and estimate its lift-to-drag coefficient, a crucial indicator of aerodynamic performance. The resulting dataset consists of wing profiles $\mathbf{x}_i$, represented by a set of 2D nodes, and the corresponding scalar lift-to-drag coefficient $\mathbf{y}_i$ for $1 \le i \le 2000$.

We selected the top 5% of shapes, based on their lift-to-drag ratio, as out-of-distribution (OOD) samples. The OOD levels were determined using the ground truth lift-to-drag ratio, where higher OOD levels correspond to more aerodynamically streamlined shapes. The training set includes shapes with lift-to-drag values ranging from 0 to 60, with anything beyond this threshold considered OOD and excluded from training. We then trained a Graph Neural Network (GNN) composed of 25 GMM (Monti et al., 2017) layers, featuring ELU activations (Clevert et al., 2015) and skip connections (He et al., 2016), to predict the lift-to-drag coefficient $\mathbf{y}_i$ from the profile $\mathbf{x}_i$, following the approach of (Remelli et al., 2020; Durasov et al., 2024a). As with previous experiments, IT³ significantly improves performance on OOD shapes and provides more accurate predictions compared to

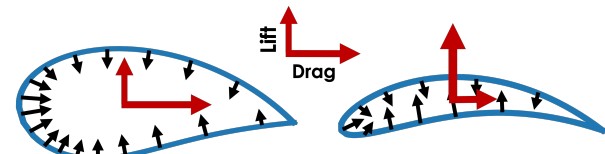

*Figure 10.* **Airfoil Samples.** Training and testing profiles (**left**) show reasonable aerodynamics, while OOD samples (**right**) feature rare, high lift-to-drag shapes. Black arrows indicate pressure, and red lines show lift and drag.

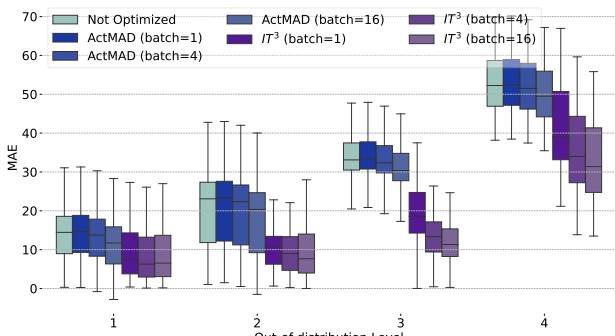

*Figure 11.* **Airfoil results on OOD shapes.** Not optimized corresponds to a single model without TTT applied. **ITTT** with [**batch=1**, **batch=4**, and **batch=16**] represents our method at different batch sizes. As the data shifts further from the training distribution, our method degrades less compared to the Not optimized and **ActMAD**, with larger batches preserving performance more effectively.

other baselines, as shown in Fig. 11.

**Cars.** As for wings, we experimented with 3D car models from a subset of the ShapeNet dataset (Chang et al., 2015), which contains car meshes suitable for CFD simulations, as depicted in Fig. 12. The experimental protocol was the same as for the wing profiles, except we used OpenFOAM (Jasak et al., 2007) to estimate drag coefficients and employed a more sophisticated network to predict them from the triangulated 3D car meshes.

To predict drag associated to a triangulated 3D car, we utilize similar model to airfoil experiments but with increased

*Table 1.* **Qualitative result for Online IT³.** Evaluation metrics for the road segmentation task (**left**), airfoils lift-to-drag prediction (**middle**), and car drag prediction (**right**). Online IT³ enhances performance compared to the original model and significantly outperforms offline IT³.

| METHOD | Corr | Comp | Quality | METHOD | MAE | METHOD | MAE |
|---|---|---|---|---|---|---|---|
| NOT OPTIMIZED | 55.7 | 44.3 | 39.5 | NOT OPTIMIZED | 38.2 | NOT OPTIMIZED | 0.501 |
| IT³ (BATCH=4) | 55.7 | 49.1 | 46.4 | IT³ (BATCH=1) | 37.6 | IT³ (BATCH=1) | 0.446 |
| IT³ (BATCH=8) | 58.1 | 52.0 | 48.5 | IT³ (BATCH=4) | 37.5 | IT³ (BATCH=2) | 0.424 |
| IT³ (BATCH=16) | 57.3 | 52.7 | 48.7 | IT³ (BATCH=16) | 37.4 | IT³ (BATCH=4) | 0.412 |
| IT³ (ONLINE) | **77.5** | **79.8** | **69.8** | IT³ (ONLINE) | **34.1** | IT³ (ONLINE) | **0.385** |

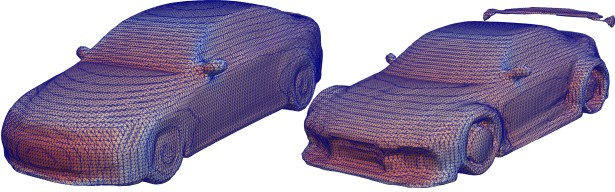

*Figure 12.* **Car Samples.** The car dataset comprises many regular vehicles (**left**) and a few streamlined ones (**right**), which we treat as being out-of-distribution. Red and blue denote high and low pressures respectively.

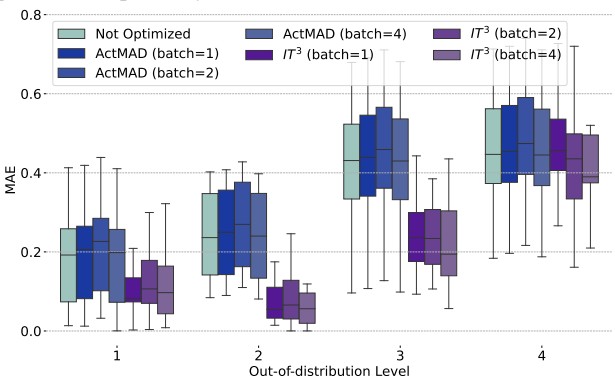

*Figure 13.* **Car results on OOD shapes.** Not optimized corresponds to a single model without TTT applied. **ITTT** with [batch=1, batch=2, and batch=4] represents our method at different batch sizes. As the data shifts further from the training distribution, our method degrades less compared to the Not optimized baseline and ActMAD, with larger batches preserving performance more effectively.

capacity. Instead of twenty five GMM layers, we use thirty five and also apply skip-connections with ELU activations. Final model is being trained for 100 epochs with Adam optimizer and $10^{-3}$ learning rate. As with airfoils experiments, IT³ significantly improves performance on OOD shapes and provides more accurate predictions compared to other baselines, as shown in Fig. 13.

## 4.6. Online IT³

We test our proposed online variation on several tasks. *Assuming a data stream online scenario rather than the previous setup.* Naturally, when the distribution remains constant (although shifted from the training distribution) we expect

superior results w.r.t. the offline setup, as our model keeps being trained on the new distribution. A more effective way to test constant adaptation over time is to use a continuously changing distribution. We test IT³ on an increasing corruption / OOD level. In all cases, the online IT³ performs significantly better than the basic anchored variation.

**Road segmentation.** Building upon our previous road segmentation experiments, we further evaluate the effectiveness of online IT³. In the online IT³ setup, OOD samples are ranked based on their mean squared error (MSE) loss when passed through the vanilla network. We begin by selecting the samples with low MSE loss, as these are closer to the training distribution given the network's strong performance on them. Gradually, we introduce samples with progressively higher MSE loss, smoothly shifting between distributions and thereby allowing the model to adapt effectively to a range of OOD samples. As in previous experiment, we use DRU-Net trained on the RoadTracer dataset as vanilla model and 890 images are sampled from Massachusetts dataset as OOD images.

Firstly, the vanilla network is tested on the Massachusetts dataset without any additional fine-tuning. We then apply online IT³ during inference to adapt the model to the OOD distribution as new data is presented. We evaluate the segmentation performance using the *Correctness*, *Completeness*, and *Quality* metrics, as described previously. Table 1 (left) summarizes the results. The application of IT³ improved the performance over the initial network and the online IT³ method significantly outperforms the offline IT³.

**Aerodynamics.** We conducted online experiments for airfoils and cars lift-to-drag prediction. We set the data stream such that OOD shapes appear with increasing aerodynamic properties, modeling a continuous domain shift in the data. As with the segmentation results, the online version significantly outperforms both the offline version and the original network, as shown in Tabs. 1 (middle and right).

## 4.7. ImageNet

ImageNet-C (Hendrycks & Dietterich, 2019) consists of ImageNet (Krizhevsky et al., 2012) test images corrupted using the same transformations as CIFAR-10/100C (Sec. 4.2). We

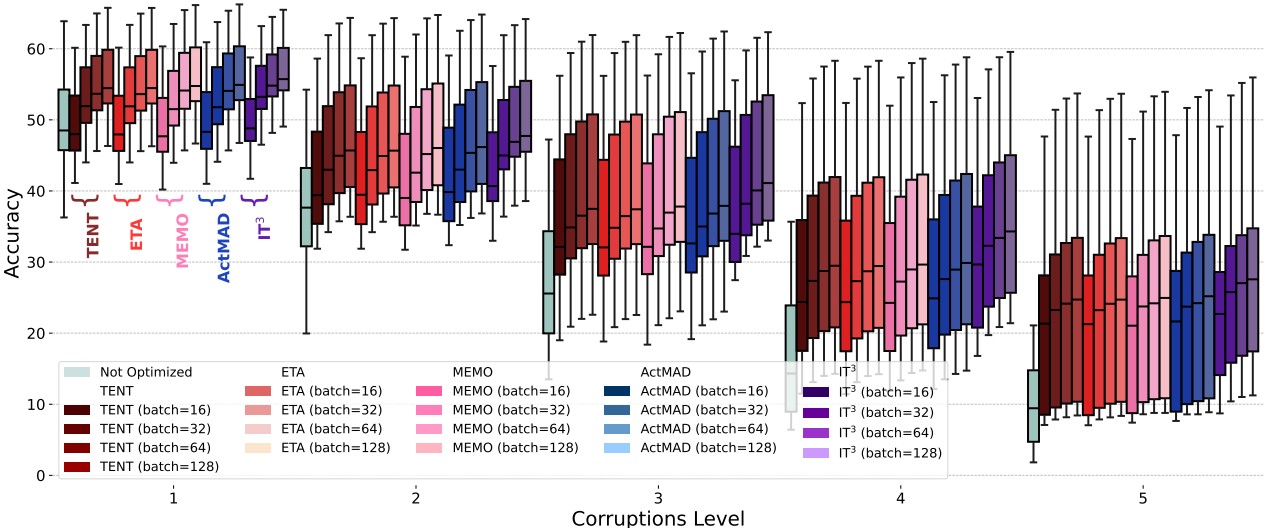

*Figure 14.* **Test accuracy (%) on ImageNet-C across 5 corruption levels.** This plot compares our method, $IT^3$, with popular adaptation approaches including TENT, ETA, MEMO, and ActMAD on the ImageNet-C dataset. $IT^3$ consistently outperforms all baselines across all corruption severity levels and batch sizes, with performance further improving at higher batch sizes.

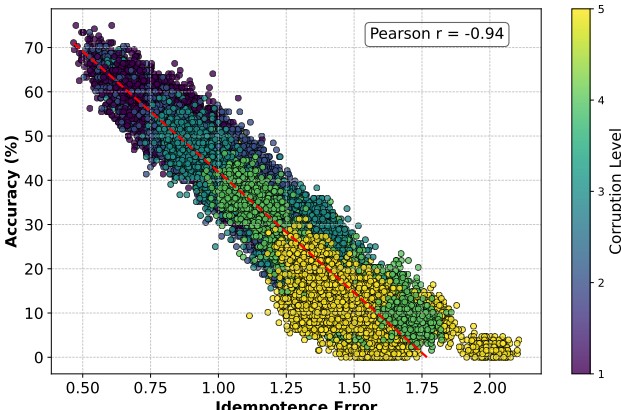

*Figure 15.* **Accuracy vs Idempotence on Corrupted ImageNet**: This plot demonstrates a strong correlation between model idempotence and performance. Each point corresponds to one inference batch, with accuracy and idempotence error computed per batch. The color indicates the corruption level from ImageNet-C for the respective batch. The Pearson correlation between idempotence error and accuracy is $-0.94$, indicating a strong negative trend.

used it for large-scale classification experiments, evaluating performance across 15 corruption types and different severity levels. For our setup, we employed a standard ResNet-18 (He et al., 2016) and followed the Pytorch training protocol (Paszke et al., 2017). In addition to our previous experiments, we included several widely used baselines on this benchmark: TENT (Wang et al., 2020), ETA (Niu et al., 2022), and MEMO (Zhang et al., 2022).

The results on ImageNet-C (each bar represents the average accuracy across 15 corruption types) are shown in Table 14. Our method outperforms all other approaches across all

corruption levels and batch sizes, and significantly surpasses the original baseline model. As previously observed, larger inference batch sizes improve performance for all methods. These results demonstrate that our approach is also effective in large-scale data scenarios.

**Accuracy vs Idempotence.** Similar to the results in Fig. 2, Fig. 15 illustrates the correlation between model performance and idempotence error. This supports the core idea of our method: optimizing idempotence during inference can improve performance. As shown, idempotence error exhibits a strong negative correlation with accuracy. This observation further reinforces the conclusions of (Shocher et al., 2024) and (Durasov et al., 2024a), which highlight that idempotence error across multiple predictions is a strong indicator of model performance—a key motivation behind our approach.

## 5. Conclusions, Limitations, and Future Work

We have proposed an approach to test-time-training relying on enforcing idempotence as new samples are being considered. This effectively handles domain shifts and method is generic. We have demonstrated that it is effective in a wide range of domains without requiring domain-specific knowledge, which sets it apart from state-of-the-art methods.

The flip side is that IT³ lacks domain expertise. In some cases, it is hard to apply IT³ to single instances without additional conditions. This is most common in domains where information within a single input is limited. Combining IT³ with domain-specific methods may remove these limitations, which we will explore in future work.

## Impact Statement

This paper introduces IT$^3$, a new approach that enables more adaptable machine-learning models by refining their predictions for out-of-distribution data at test time. Through this on-the-fly adaptation, IT$^3$ can reduce training costs, data requirements, and model size, making advanced AI methods more broadly accessible. The technique thus offers a step toward more efficient, flexible deployment of deep learning in real-world scenarios, where conditions often shift beyond the original training domain.

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

## A. Inference Time Comparison

Our method typically requires only 1–3 optimization steps, keeping the overall cost comparable to other well-known TTT methods. Below, we provide a comparison of inference times on out-of-distribution data for three approaches: the base model without optimization, the state-of-the-art TTT method ActMAD, and IT³. As shown, while our method introduces no significant overhead compared to ActMAD, it remains computationally efficient while achieving substantial improvements in performance. A similar observation can be made about memory consumption, as reported in Tab. 5 for the case of batch size 128 on ImageNet-C, showing peak memory reserved (in GB) using `torch.cuda.max_memory_reserved()`.

*Table 2.* Inference Time Comparison (OOD Airfoils)

| Method | Base Model | ActMAD | IT³ |
|---|---|---|---|
| Inference Time ($\downarrow$) | $1\times$ | $3\times$ | $4\times$ |

*Table 3.* Inference Time Comparison (OOD Cars)

| Method | Base Model | ActMAD | IT³ |
|---|---|---|---|
| Inference Time ($\downarrow$) | $1\times$ | $4\times$ | $5\times$ |

*Table 4.* Inference Time Comparison (OOD Roads)

| Method | BASE | ActMAD | IT³ |
|---|---|---|---|
| Inference Time ($\downarrow$) | $1\times$ | $4.5\times$ | $6\times$ |

*Table 5.* Memory Consumption (OOD ImagetNet), GPU Gb

| Method | BASE | TENT | MEMO | ETA | ActMAD | IT³ |
|---|---|---|---|---|---|---|
| Memory ($\downarrow$) | 4.5 | 4.8 | 13.5 | 4.9 | 7.2 | 7.4 |

## B. Extended Discussion: Relating IT³ to IGN

In this section, we elaborate on how IT³ generalizes the *projection principle* of Idempotent Generative Networks (IGN) to a supervised test-time training setting. We show that both approaches use *idempotence*—repeated applications of the network function should yield the same result—as a way to "project" off-manifold inputs onto a learned manifold of valid data. While IGN enforces idempotence directly on all possible inputs, IT³ enforces it primarily on training data but adapts on-the-fly at test time to handle out-of-distribution (OOD) samples.

**The Projection Principle in IGN.**

IGN (Shocher et al., 2024) learns $g_\theta : \mathcal{Z} \to \mathcal{X}$, mapping from a *source* distribution $\mathcal{P}_z$ (e.g. Gaussian noise) to a *target* distribution $\mathcal{P}_x \subset \mathcal{X}$ (e.g. natural images). It imposes:

$$g_\theta\big(g_\theta(z)\big) \; = \; g_\theta(z) \quad \forall\, z \in \mathcal{Z},$$

so a second application of $g_\theta$ makes no change. This idempotence implies that once an off-manifold $z$ is mapped to $g_\theta(z)$, it must already lie on the manifold $\{x : g_\theta(x) = x\}$. In effect,

$$z \; \mapsto \; g_\theta(z) \; \in \; \Big\{x : \, g_\theta(x) = x\Big\}.$$

One can interpret this as a *projection*: a drift or energy measure $\delta_\theta(x) = \| g_\theta(x) - x \|$ vanishes ($\delta_\theta(x) = 0$) if and only if $x$ already lies on that manifold. Enforcing $g_\theta(g_\theta(z)) = g_\theta(z)$ ensures $\delta_\theta(g_\theta(z)) = 0$. Hence, after one forward pass, the corrupted or noisy input is "pulled" onto the learned data manifold, and repeated applications do not alter it further.

**Idempotence in IT³: Pairwise Function.**

IT³ deals with a supervised model

$$f_\theta : \mathcal{X} \times \mathcal{Y} \; \to \; \mathcal{Y},$$

where $\mathbf{x} \in \mathcal{X}$ is an input and $\mathbf{y} \in \mathcal{Y}$ its desired output. The training set $\{(\mathbf{x}_i, \mathbf{y}_i)\}$ spans an in-distribution $\mathcal{P}_{x,y}$. During training, IT³ enforces:

1. $f_\theta(\mathbf{x}, \mathbf{y}) = \mathbf{y}$ for training pairs $(\mathbf{x}, \mathbf{y})$. Thus, each real pair is a fixed point.

2. $f_\theta(\mathbf{x}, \mathbf{0}) \approx \mathbf{y}$, using a "neutral" label $\mathbf{0}$ to predict $\mathbf{y}$.

Combining these yields:

$$f_\theta\Big(\mathbf{x},\ f_\theta(\mathbf{x}, \mathbf{0})\Big)\ =\ f_\theta(\mathbf{x}, \mathbf{0}),$$

an idempotence condition parallel to IGN's $g_\theta(g_\theta(z)) = g_\theta(z)$. One may define a drift-like measure

$$\Delta_\theta(\mathbf{x})\ =\ \Big\| f_\theta\big(\mathbf{x},\ f_\theta(\mathbf{x}, \mathbf{0})\big)\ -\ f_\theta(\mathbf{x}, \mathbf{0})\Big\|.$$

When $\mathbf{x}$ is in-distribution, training makes $\Delta_\theta(\mathbf{x}) = 0$. If $\mathbf{x}$ is OOD, $\Delta_\theta(\mathbf{x}) > 0$ initially. **Test-time adaptation** then updates $\theta$ on-the-fly to push $\Delta_\theta(\mathbf{x})$ closer to zero, thereby restoring idempotence.

### Subtlety: Internal Representations Projection

A natural question arises: *If the OOD variable* $\mathbf{x}$ *itself stays fixed, how can* $(\mathbf{x}, \mathbf{y})$ *become "on distribution"?* The answer is that, inside the network layers, $\mathbf{x}$ and $\mathbf{y}$ jointly produce hidden representations. Although $\mathbf{x}$ does not physically change, the *way* $\mathbf{x}$ participates in the representation *does* change once $\mathbf{y}$ and the model parameters $\theta$ are updated. Thus, even if $\mathbf{x}$ is not from the training distribution, the pair $(\mathbf{x}, \widehat{\mathbf{y}})$ can enter a region of representation space that *matches* valid training pairs. Formally, each layer of $f_\theta$ has activations that depend on both $\mathbf{x}$ and $\mathbf{y}$. By adjusting $\theta$ (but freezing the outer function call) so that

$$f_\theta\Big(\mathbf{x},\ f_\theta(\mathbf{x}, \mathbf{0})\Big) = f_\theta(\mathbf{x}, \mathbf{0}),$$

we effectively *project* $(\mathbf{x}, \mathbf{0})$ into the manifold $\{(\mathbf{x}, \mathbf{y}) : f_\theta(\mathbf{x}, \mathbf{y}) = \mathbf{y}\}$ *within* the network's internal representation. Hence, even though $\mathbf{x}$ remains the same, the final pair $(\mathbf{x}, \widehat{\mathbf{y}})$ is "valid" in the sense that repeated applications are stable.

### Conclusion: IT³ Also "Projects" Off-Manifold Pairs.

In IGN, once trained, any input $z \in \mathcal{Z}$ maps to $g_\theta(z)$ on the real-image manifold ($g_\theta(x) = x$). In IT³, a new OOD pair $(\mathbf{x}, \mathbf{0})$ is adapted so that $(\mathbf{x}, f_\theta(\mathbf{x}, \mathbf{0}))$ belongs to the set $\{(\mathbf{x}, \mathbf{y}) : f_\theta(\mathbf{x}, \mathbf{y}) = \mathbf{y}\}$. From an internal representation viewpoint, this *pulls* the OOD pair onto the manifold of valid $(\mathbf{x}, \mathbf{y})$ relations. Thus, IT³ *extends* IGN's core idea of "learned idempotent projection" to a supervised test-time training paradigm. Despite leaving $\mathbf{x}$ intact, the final output indeed corresponds to a consistent $(\mathbf{x}, \mathbf{y})$-pair on the model's manifold, much like IGN pulls corrupted noise into the real-data manifold.

## Appendix C: Detailed Elaboration: Relation between Adaptation and Idempotence

In this appendix, we provide a detailed explanation of the rationale behind the idempotence loss used in IT³. Our aim is to demonstrate how the discrepancy between recursive model outputs quantifies the out-of-distribution (OOD) uncertainty and why actively minimizing this discrepancy drives the model toward idempotence with respect to its auxiliary input.

### 1. Training via the ZigZag Approach

Following (Durasov et al., 2024a), the network is modified to accept an auxiliary input. During training, for each training pair $(x, y)$, the model $f$ is trained to satisfy:

$$f(x, 0) \approx y \quad \text{and} \quad f(x, y) \approx y. \tag{8}$$

This is achieved by minimizing a composite loss:

$$L_{\text{train}} = \|f(x, 0) - y\| + \|f(x, y) - y\|. \tag{9}$$

This training strategy ensures that incorporating the auxiliary input (either the true label or a "don't know" signal) does not hurt the primary task performance while establishing a consistency property in the network.

## 2. Test-Time Recursive Inference and Uncertainty Measurement

At test time, the network is applied recursively:

$$y_0 = f(x, 0), \tag{10}$$
$$y_1 = f(x, y_0). \tag{11}$$

We define the uncertainty loss as:

$$L_{\text{IT}^3}(x) = \|y_1 - y_0\|. \tag{12}$$

The rationale is as follows:

1. If $x$ is in-distribution, then $y_0 \approx y$, and since the network is trained so that $f(x, y) \approx y$, we have $y_1 \approx y_0$. Therefore, $L_{\text{IT}^3}(x)$ is small.

2. If $x$ is OOD, then $y_0$ is unlikely to approximate the true label. In this case, the pair $(x, y_0)$ is not a valid input as per training, leading $y_1$ to be unpredictable and significantly different from $y_0$, resulting in a large $L_{\text{IT}^3}(x)$.

Thus, the magnitude of $\|y_1 - y_0\|$ serves as a proxy for prediction certainty.

## 3. From Uncertainty to Idempotence

We now elaborate on how the uncertainty loss translates into enforcing idempotence of $f(x, \cdot)$ for a given $x$. Recall that a function is idempotent if

$$f(f(x)) = f(x). \tag{13}$$

For our setting, consider the function $g(y) = f(x, y)$. The desired idempotence condition becomes:

$$g(g(0)) = g(0), \tag{14}$$

or equivalently,

$$f(x, f(x, 0)) = f(x, 0). \tag{15}$$

Thus, we can rewrite the uncertainty loss as:

$$L_{\text{IT}^3}(x) = \|f(x, f(x, 0)) - f(x, 0)\|. \tag{16}$$

Minimizing $L_{\text{IT}^3}(x)$ drives the network toward the condition that repeated application of $f(x, \cdot)$ does not change the output. When $L_{\text{IT}^3}(x) = 0$, the function $f(x, \cdot)$ is idempotent given the input $x$. This self-consistency indicates that the network's output is aligned with the in-distribution manifold.

## 4. Addressing Optimization Challenges

Minimizing the idempotence loss is not trivial. Naively reducing $L_{\text{IT}^3}$ can lead to pitfalls such as reinforcing an erroneous prediction $y_0$. As discussed in (Shocher et al., 2024), directly optimizing the idempotence loss induces two gradient pathways:

1. A desirable pathway that updates $f(x, 0)$ toward the correct in-distribution manifold.

2. An undesirable pathway that may cause the manifold to expand, thereby including an incorrect $f(x, 0)$.

To counteract the latter, our method avoids passing gradients through the second application of $f$ by using a frozen copy of the network (or updating it via an exponential moving average). Specifically, we compute:

$$y_1 = F(x, f(x, 0)), \tag{17}$$

and redefine the loss as:

$$L_{\text{IT}^3} = \|F(x, f(x, 0)) - f(x, 0)\|. \tag{18}$$

This decoupling ensures that only the first prediction $y_0$ is adapted during test-time training, thus preventing error reinforcement and ensuring that the minimization of the loss indeed pulls $f(x, \cdot)$ toward idempotence.

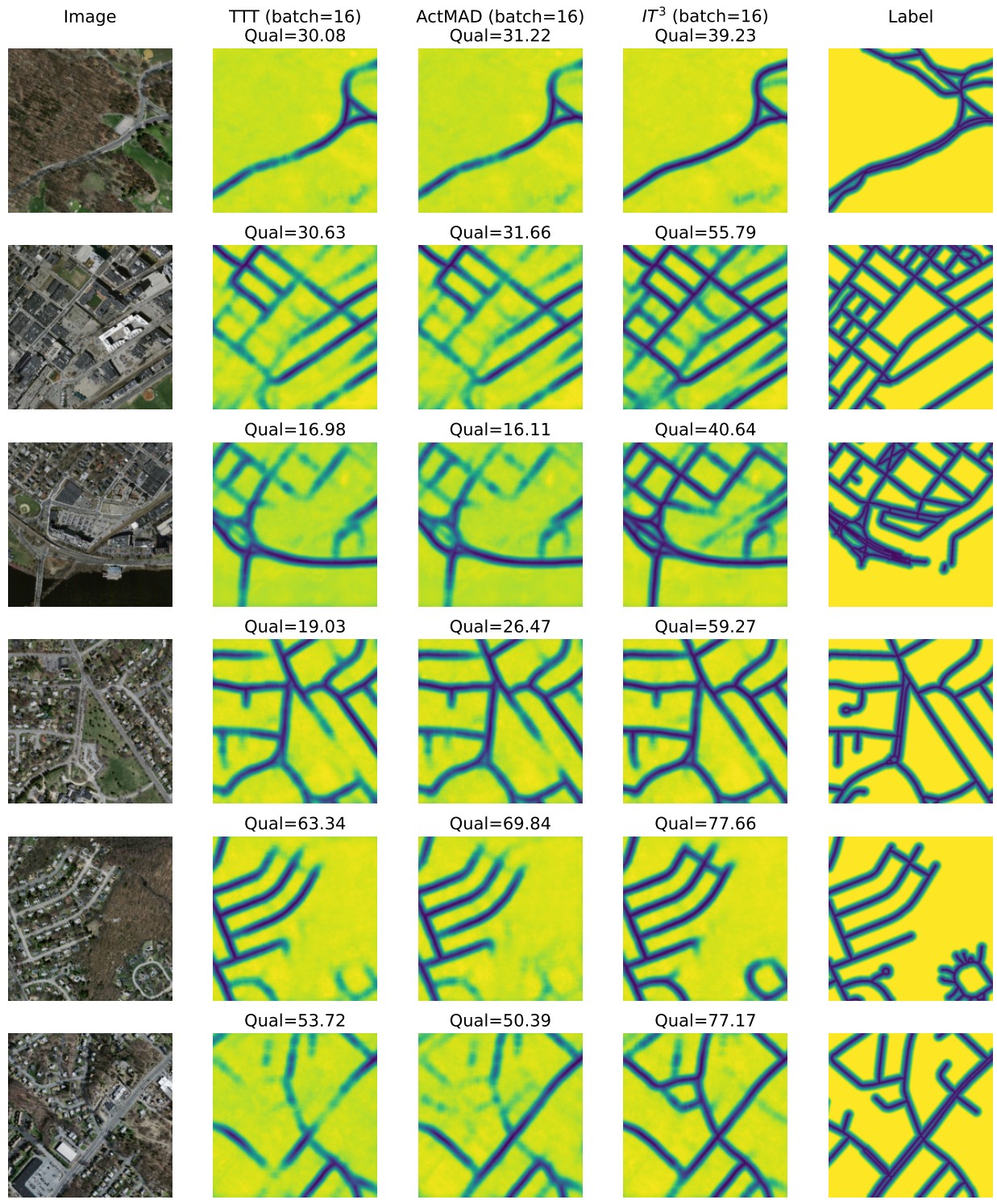

*Figure 16.* **Comparison of different Test-Time Training methods on segmentation tasks.** This plot shows the predictions of various TTT methods on an aerial segmentation task. As can be seen, our approach consistently enhances prediction quality and outperforms other methods.

## 5. Summary

By minimizing the loss

$$L_{\text{IT}^3} = \|F(x, f(x, 0)) - f(x, 0)\|, \tag{19}$$

we enforce the condition

$$f(x, f(x, 0)) \approx f(x, 0), \tag{20}$$

i.e., $f(x, \cdot)$ becomes idempotent given the input $x$. This idempotence is a critical indicator that the input is aligned with the training distribution. When $L_{\text{IT}^3}$ is small, the prediction is self-consistent and reliable; when it is large, it signals that the input is likely OOD. Consequently, actively minimizing this loss through test-time training refines the network's prediction and enhances its robustness to distribution shifts.

For further details on related projection perspectives, please refer to Appendix B.

