# OpenReview forum: "IT$^3$: Idempotent Test-Time Training"
_ICML.cc/2025/Conference — ICML 2025 poster_

### Official Review · Reviewer_Viky · 2025-03-11

**Overall Recommendation:** 3

**Summary:**

This paper introduces IT3, a test-time training method that leverages idempotence to adapt model weights on-the-fly without requiring domain-specific auxiliary tasks. By enforcing that repeated applications of the model yield the same output, IT3 effectively projects out-of-distribution inputs onto the training data manifold, achieving improved robustness across diverse tasks and domains.

**Claims And Evidence:**

Most claims are supported by extensive experimental evidence, but the theoretical justification of idempotence as a projection mechanism of the test-time training process are less substantiated.

**Essential References Not Discussed:**

No essential references appear to be missing beyond those already mentioned.

**Experimental Designs Or Analyses:**

I examined the experimental design, and a key concern is the limited set of baselines: while the paper occasionally compares IT³ to methods like ActMAD, it does not consistently benchmark against TTT++ across all tasks with matching batch sizes and architectures.


TTT++: When Does Self-Supervised Test-Time Training Fail or Thrive?, NeurIPS 2021

**Methods And Evaluation Criteria:**

While the proposed methods align conceptually with test-time training for handling distribution shifts, the experiments primarily involve smaller datasets and omit standard OOD benchmarks like WILDS, leaving broader applicability and real-world relevance underexamined.

Wilds: A benchmark of in-the-wild distribution shifts. ICML, 2021.

**Other Comments Or Suggestions:**

1. Provide more detail on using zero as a neutral signal in regression tasks, where zero may also be a valid label.
2. Briefly discuss tuning hyperparameters (e.g., optimization steps, learning rate) for test-time adaptation to guide practical deployment.

**Other Strengths And Weaknesses:**

Strengths:

1. The approach leverages a general idempotence principle rather than domain-specific tasks, making it broadly applicable.
2. It consistently demonstrates improved performance on diverse tasks (e.g., tabular data, image classification, aerodynamics).
3. The method is relatively straightforward to plug into existing architectures and does not require extensive auxiliary supervision.

Weaknesses:

1. If the model’s initial prediction is incorrect, even with the frozen reference model, the iterative optimization can magnify the error signal.
2. The tasks used in experiments are relatively small-scale or synthetic, leaving questions about real-world robustness.

**Questions For Authors:**

1. Could you clarify how you prevent confusion between the neutral zero and legitimate zero labels in regression tasks, and whether an alternative placeholder value might be more appropriate?

2. How does IT³ scale in terms of memory and runtime when model sizes grow, and have you tested it on larger networks (e.g., ImageNet-scale) to confirm its practicality?

3. When the initial prediction for an OOD sample is substantially off, what mechanisms (if any) prevent the model’s iterative updates from further reinforcing incorrect outputs?

**Relation To Broader Scientific Literature:**

The paper extends the emerging line of work on test-time adaptation (e.g., TTT, ActMAD) by introducing idempotence as a unifying principle that obviates the need for domain-specific auxiliary tasks.

**Theoretical Claims:**

The paper’s theoretical justification relies largely on intuitive arguments relating idempotence to test-time adaptation, rather than providing formal proofs. While the extended discussion references prior work (e.g., IGN), it does not include rigorous derivations confirming that test-time adaptation preserves idempotence on OOD samples, leaving this point insufficiently substantiated.

---

> ### Author Rebuttal · Authors · 2025-04-01
>
> We appreciate the reviewer’s insightful comments. We address the reviewer's concerns below and will revise the paper accordingly.
>
> &nbsp;
>
> ### **"The experiments primarily involve smaller datasets and omit standard OOD benchmarks like WILDS, leaving broader applicability and real-world relevance underexamined." / "limited set of baselines" / "The tasks used in experiments are relatively small-scale"How does IT³ scale in terms of memory and runtime when model sizes grow, and have you tested it on larger networks (e.g., ImageNet-scale) to confirm its practicality?"**
>
> Following the reviewer’s suggestion, we conducted additional experiments on **ImageNet-C** and compared also to TTA baselines:   **TENT, ETA and MEMO** across various batch sizes. The results are **[PRESENTED IN THIS PLOT](https://imgur.com/a/imagenet-corruptions-robustness-bGaru0u)**, which will be included in the final version of the paper. For each method and severity level, we present bars composed of 15 accuracy values—one for each corruption type. Our method consistently outperformed with a 3–5% improvement. While WILDS  is clearly relevant, prior TTT works did not evaluate on it and we missed it. In the limited rebuttal time we had, we prioritized ImageNet-C and additional baselines experiments. We will test on standard OOD benchmarks when revising the paper.
>
> Runtime analysis can be found in Appendix A. When revising, we will add analysis of memory consumption to all cases as well.
> One case for batch size 128 ImageNet-C in GB:
>
> TENT: 4.8
> ETA: 4.9
> MEMO: 13.5  (3 augmentations)
> ActMAD: 7.2
> IT$^3$: 7.4
> Vanilla Model: 4.5
>
> &nbsp;
>
> ### **"... does not include rigorous derivations confirming that test-time adaptation preserves idempotence on OOD samples."**
>
> Thank you for identifying this important issue. We acknowledge that the paper lacks a clear logical flow connecting idempotence to OOD adaptation. We provide here **[A SKETCHED LOGICAL CHAIN](https://imgur.com/XwHPhbc)** explaining exactly this. While not a formal proof,  it is a rigorous description of this relation. This will be combined into the introduction in the paper.  A detailed extended version will be added to the paper as appendix C. A very brief summary is below:
>
> 1. When using ZigZag approach to training, the discrepancy between recursive calls of the model, $|| y_1 - y_0 ||$, is highly correlated to how far out-of-distribution (OOD) an input is.
> 2. Thus, we can use $|| y_1 - y_0 ||$ as a loss, and reduce 'OODness'.
> 3. By doing this, we make the network more idempotent: $L_{\mathrm{IT^3}} = || y_1 - y_0 || = || f(x, f(x,0)) - f(x,0) ||.$
> 4.  But properly minimizing $L_{\mathrm{IT^3}}$ is non trivial, so we employ the approach from IGN (Shocher et al.)
> 5. This can be understood as a projection onto a distribution. (Appendix B).
>
> To empirically support this claim, we also analyzed the relationship between corruption severity and idempotence loss on ImageNet-C. The results are shown **[IN THIS LINK](https://imgur.com/a/idempotence-vs-imagenet-performance-ym0McKq)**. Each point represents a batch of data, with the x-axis showing the computed idempotence loss and the y-axis indicating classification accuracy. As expected, higher severity levels correspond to greater idempotence loss. Notably, we observed a strong negative correlation between idempotence loss and performance (Pearson correlation: –0.94).
>
> &nbsp;
>
> ### **"Briefly discuss tuning hyperparameters (e.g., optimization steps, learning rate) for test-time adaptation to guide practical deployment.**"
>
> Thank you for this important comment. We will expand appendix A to include implementation details. Code will be released upon acceptance.
>
> &nbsp;
>
> ### **"Could you clarify how you prevent confusion between the neutral zero and legitimate zero labels?"**
>
> We thank the reviewer for their important comment. The special 0 notation used does not represent an actual zero. Instead, it is a unique signal with the same dimensions as the labels, specifically chosen to differentiate it from actual label values. Our approach builds upon the ZigZag method from [1], where the choice of this signal is extensively discussed and justified. We understand that this notation may appear confusing, so we will revise the paper to include a clearer explanation to avoid any ambiguity.
>
> &nbsp;
>
> ### "**If the model’s initial prediction is incorrect, even with the frozen reference model, the iterative optimization can magnify the error signal." / "When the initial prediction for an OOD sample is substantially off, what mechanisms prevent... reinforcing incorrect outputs?"**
>
> This is a valid point. Even with the frozen network trick, it is not impossible for such event to occur. Empirically, such cases are rare in comparison to cases where the TTT session improved the prediction. The number of TTT iterations is very limited. The network is reset after each instance so no long term damage can really accumulate.

---

### Official Review · Reviewer_tgc2 · 2025-03-13

**Overall Recommendation:** 3

**Summary:**

This paper proposes a test-time adaptation (TTT) method named Idempotent Test-Time Training (IT3), that enforces idempotence to the model. Here, idempotence indicates that repeating a function to end in a stationary point (or fixed point). Specifically, IT3 predicts the output with a given input and randomly shown label, where at test-time the model is trained to minimize the ‘first prediction with null label as input’ and the successive model’s prediction with the self-predicted label. The method is tested across multiple domains and architectures, showing promising results.

----

**POST REBUTTAL**

For a short summary, I will raise the score to weak acceptance, but would like to highlight a small concern that I have.

Most of my concerns are addressed, leaving only one concern. "Why does this proposed method improve OOD robustness"?.

From a personal perspective as a reviewer (or a reader), the rebuttal did not fully clarify my concern and question. I think Idempotent somewhat makes sense to improve OOD robustness, as OOD makes the model struggle more to have a consistent output. However, it is also somewhat unclear why making Idempotent consistent on the in-distribution sample improves OOD robustness. I think this question is raised by other reviewers as well (e.g., WBsB).

But also from the author's perspective (I have also done some research on this domain), OOD robustness is hard to prove and sometimes an empirical science. So if the OOD robustness claim is toned down a little, it would be much helpful.

**Claims And Evidence:**

While most of the claims are reasonable, I believe the key idea and engineering technique should be clarified, namely the justification of generalization ability of IT3 and the use of EMA.

**Lack of justification for IT3’s generalization ability to out-of-distribution (OOD) data.** \
While the paper presents IT3 as an effective method for handling OOD data scenarios, the justification for why enforcing idempotence leads to better OOD generalization. The intuition behind this connection needs to be more explicitly stated, and additional ablation studies could help clarify why this approach is effective beyond empirical results.

**Unclear contribution of exponential moving average (EMA) to the performance gains.** \
The role of EMA in improving performance is not well justified. While it is used to stabilize adaptation in the online setting, the underlying reasons for its significant impact on performance are not well explored. I think this result is quite interesting and justifying well will improve the paper's claim.

**Essential References Not Discussed:**

The paper presented a good related work section.

**Experimental Designs Or Analyses:**

I have checked the soundness and validity of experimental design and analysis, which seems correct. I am not claiming that the experiment is sufficient rather claiming that the presented experiment seems correct (see other parts to find the weakness).

**Methods And Evaluation Criteria:**

The evaluation criteria are correct for the proposed method. But I do believe the paper needs more baseline and experiments to fully support the paper's claim.

**Limited large-scale experiments.** \
The experiments primarily focus on CIFAR-10 (for image) and other relatively small-scale datasets. To convincingly demonstrate scalability, it would be beneficial to include results from larger datasets such as ImageNet. If it is hard to run ImageNet during the rebuttal it would be great if the author can run ImageNet subset. Without this, it remains unclear how well IT3 scales to more complex real-world settings.

**Missing baselines.** \
The author compares IT3 with some TTT methods but lacks comparison with augmentation-based baselines. While it is true that augmentation-based baselines require domain-specific knowledge, I believe most of the real-world applications actually know which augmentation to use at train/test time. In this regard, I believe it is great to compare with such baselines like "MEMO: Test Time Robustness via Adaptation and Augmentation." [1] Additionally, batch normalization-based approaches such as "Tent: Fully Test-time Adaptation by Entropy Minimization" [2] should be included, especially since batch normalization layers can be updated without requiring domain knowledge. BN-based method can be used without architecture modification, but this method requires architecture modification. Also,  comparison with "single-point BN" used in MEMO would also be relevant.

**Computational overhead and fairness in comparisons.** \
IT3 requires maintaining two networks (original and EMA), which increases memory and computational requirements. This could lead to unfair comparisons with single-network baselines that do not require additional storage for EMA parameters. A fairer comparison should analyze compute overhead and consider normalized performance metrics that account for increased memory usage (or parameter count).

Reference\
[1] MEMO: Test Time Robustness via Adaptation and Augmentation, NeurIPS 2022\
[2] Tent: Fully Test-time Adaptation by Entropy Minimization, ICLR 2021

**Other Comments Or Suggestions:**

**Impact of label dimension (|y|) on performance should be studied.** \
Since the IT3 framework explicitly requires designing a network where y is part of the input, it is important to analyze how performance varies as the number of classes increases. The current experiments mainly focus on small-class problems like CIFAR-10 (in image), but it is unclear whether the method remains effective when dealing with datasets with higher label dimensionality. I think showing more high-dimensional (label) domain in the image domain will be great, as most readers and reviewers are more familiar with image baselines.

**Other Strengths And Weaknesses:**

For strength, I think the overall paper is well written and clearly discusses the difference between test-time adaptation and test-time training. Also, while I still do think the domain agnostic needs comparison with domain specific methods (e.g., augmentation-based), the paper did a great job to consider multiple domains (e.g., Tabular) that lack domain specific knowledge.

Other weakness are presented in other sections. I think the main weakness is the justification of Idempotent for OOD generalization. I kindly ask the authors to address this issue during the rebuttal.

**Questions For Authors:**

All questions are asked in other sections.

**Relation To Broader Scientific Literature:**

I think the main contribution of this paper is introducing a new idea (i.e., Idempotence) to test-time training which is quite interesting. If the author could well connect the reason why Idempotence helps generalization to OOD dataset, it would be very helpful.

**Theoretical Claims:**

There is no theoretical claim in the paper.

---

> ### Author Rebuttal · Authors · 2025-04-01
>
> We sincerely thank the reviewer for their detailed and constructive feedback. We address the reviewer's concerns below and will revise the paper accordingly.
>
> &nbsp;
>
> ### **"Lack of justification for IT3’s generalization ability to out-of-distribution (OOD) data. While the paper presents IT3 as an effective method for handling OOD data scenarios, the justification for why enforcing idempotence leads to better OOD generalization. The intuition behind this connection needs to be more explicitly stated, and additional ablation studies could help clarify why this approach is effective beyond empirical results."**
>
> Thank you for identifying this important issue. We acknowledge that the paper lacks a clear logical flow connecting idempotence to OOD adaptation. We provide here **[A SKETCHED LOGICAL CHAIN](https://imgur.com/a/imagenet-corruptions-robustness-bGaru0u)** explaining exactly this. This will be combined into the introduction in the paper. A detailed extended version will be added to the paper as appendix C. A very brief summary is below:
>
> 1. When using ZigZag approach to training, the discrepancy between recursive calls of the model, $|| y_1 - y_0 ||$, is highly correlated to how far out-of-distribution (OOD) an input is.
> 2. Thus, we can use $|| y_1 - y_0 ||$ as a loss, and reduce 'OODness'.
> 3. By doing this, we make the network more idempotent: $L_{\mathrm{IT^3}} = || y_1 - y_0 || = || f(x, f(x,0)) - f(x,0) ||.$
> 4.  But properly minimizing $L_{\mathrm{IT^3}}$ is non trivial, so we employ the approach from IGN (Shocher et al.)
> 5. This can be understood as a projection onto a distribution. (Appendix B).
>
>
> To empirically support this claim, we also analyzed the relationship between corruption severity and idempotence loss on ImageNet-C. The results are shown **[IN THIS LINK](https://imgur.com/a/idempotence-vs-imagenet-performance-ym0McKq)**. Each point represents a batch of data, with the x-axis showing the computed idempotence loss and the y-axis indicating classification accuracy. As expected, higher severity levels correspond to greater idempotence loss. Notably, we observed a strong negative correlation between idempotence loss and performance (Pearson correlation: –0.94).
>
> &nbsp;
>
> ### **"Unclear contribution of exponential moving average (EMA) to the performance gains."**
> The online version, as described in sec. 3.3 differs from the base version because they aim at different scenarios. In the base version,  the assumption is that each input is a separate test and has no correlation or information about other inputs. Thus the network weights are reset back to the state they were at the end of the pre-training phase. In contrast, the online version is intended for a continual learning data stream with correlation and somewhat smooth transitioning between inputs. In this case, instead of resetting after each input, we leave the weights updated from the previous inputs.
>
> Since the data keeps shifting in this scenario, over time, $f_{\theta}$ may diverge far from $F$, making it irrelevant. Instead, we need an anchor that is influenced by a reasonable amount of data, yet evolves over time. Our solution is to replace $F$ with an Exponential Moving Average (EMA) of the model. This means $f_{\text{EMA}}$ is a smoothed version of $f_{\theta}$ over time.
>
> Note that the online-IT$^3$ results, depicted in Table. 1 achieve higher scores than the base model. This is due to the inherent difference between the base and the online scenarios. A correlated data stream allows aggregating  information during test-time and therefore expected to exploit it and perform better.
>
> &nbsp;
>
> ### **"Limited large-scale experiments... it would be beneficial to include results from larger datasets such as ImageNet." / "I think showing more high-dimensional (label) domain in the image domain will be great." / "Missing baselines... lacks comparison with augmentation-based baselines."**
>
> Following the reviewer’s suggestion, we conducted additional experiments on **ImageNet-C** and compared also to TTA baselines:   **TENT, ETA and MEMO** across various batch sizes. The results are **[PRESENTED IN THIS PLOT](https://imgur.com/a/imagenet-corruptions-eI7JNfr)**, which will be included in the final version of the paper. For each method and severity level, we present bars composed of 15 accuracy values—one for each corruption type. Our method consistently outperformed with a 3–5% improvement.
>
> &nbsp;
>
> ### **"Computational overhead and fairness in comparisons... A fairer comparison should analyze compute overhead and consider normalized performance metrics that account for increased memory usage (or parameter count)."**
>
> Runtime analysis can be found in Appendix A. We will add analysis of memory consumption to all cases as well.
> One case for batch size 128 ImageNet-C, peak memory reserved with torch.cuda.max_memory_reserved() in GB:
>
> TENT: 4.8
> ETA: 4.9
> MEMO: 13.5  (3 augmentations)
> ActMAD: 7.2
> IT$^3$: 7.4
> Vanilla Model: 4.5

---

### Official Review · Reviewer_4J9m · 2025-03-13

**Overall Recommendation:** 3

**Summary:**

This paper proposes a novel test-time learning objective based on idempotent learning. The pipeline is easy to use, appears to be task-agnostic and model-agnostic, and is thus more versatile than previous TTA solutions. Experiments across various tasks demonstrate the effectiveness of the proposed method. My detailed comments are as follows.

**Claims And Evidence:**

The claim "enables on-the-fly adaptation to distribution shifts using only the current test instance, without any auxiliary task design." needs to be further clarified. Please refer to my Questions part.

**Essential References Not Discussed:**

NO

**Experimental Designs Or Analyses:**

2. For classification results, more comparisons with advanced fully test-time adaptation (TTA) methods, such as TENT, EATA, and DEYO, as well as evaluations on larger datasets like ImageNet-C, would strengthen the justification for the proposed method’s superiority.

3. For segmentation results, could the authors include additional evaluations on commonly used benchmarks such as KITTI-C and nuScenes? This would enable a fairer comparison, as the dataset used in this paper has not been widely adopted in previous works.

**Methods And Evaluation Criteria:**

Yes, The designs make sense.

**Other Comments Or Suggestions:**

5. The error bars in Figures 3, 4, and 8 are difficult to recognize and distinguish, particularly for color vision-deficient readers. Additionally, could the authors provide more qualitative results instead of bar charts? The absolute differences in bar heights are not intuitive to interpret.

**Other Strengths And Weaknesses:**

++Pros

The idea of leveraging idempotent learning for test-time adaptation (TTA) is interesting. This learning pipeline is not restricted to specific model architectures (e.g., MLP, CNN, Transformer) or specific tasks (e.g., classification, regression, segmentation).

The overall method is technically sound, and simple yet effective.

Extensive experiments on classification, tabular data regression, age prediction, and road segmentation demonstrate the promise of the proposed approach.

The paper is well-written and easy to follow.

**Questions For Authors:**

1 The authors claim that their method does not require designing additional auxiliary tasks or using extra data, making the adaptation on-the-fly. However, from my perspective, enforcing idempotency during training can also be considered an auxiliary task, and it also requires access to training data for supervised learning. Could the authors clarify this further?

4 Could the authors provide a computational complexity analysis comparing their method with existing baselines?

**Relation To Broader Scientific Literature:**

The problem studied is a fundamental challenge in machine learning with great potential for practical applications.

**Theoretical Claims:**

No theoretical claims.

---

> ### Author Rebuttal · Authors · 2025-04-01
>
> We sincerely thank the reviewer for their thoughtful and encouraging feedback. We address the reviewer's concerns below and will revise the paper accordingly.
>
> &nbsp;
>
> ### **"The authors claim that their method does not require designing additional auxiliary tasks or using extra data, making the adaptation on-the-fly. However, from my perspective, enforcing idempotency during training can also be considered an auxiliary task,...**
>
> Indeed, enforcing idempotence can be seen as an auxiliary task. However, it is not a domain-specific one. This is in contrast to approaches that use auxiliary tasks that are specific for data-type. Take for instance Sun et al., or Gandelsman et al. Their visual auxiliary tasks cannot be used on other types of data. Some others are architecture-dependent e.g., ActMAD (Mirza et al., 2023). Contrary to existing work, the IT$^3$ mechanism can be used out of the box for any task (we demonstrate image classification, aerodynamics prediction, aerial segmentation and tabular data) and any architecture (MLPs, CNNs, GNNs).
>
> &nbsp;
>
> ### **...and it also requires access to training data for supervised learning. Could the authors clarify this further?"**
>
> During the pre-training phase, IT$^3$ uses the training data like any other supervised-learning method. At test time, the challenge is defined so that there is absolutely no access to the training data. At any point, the only thing given is the current test input.
>
> &nbsp;
>
> ### **"For classification results, more comparisons with advanced fully test-time adaptation (TTA) methods, such as TENT, ETA, and DEYO, as well as evaluations on larger datasets like ImageNet-C, would strengthen the justification for the proposed method’s superiority."**
>
> Following the reviewer’s suggestion, we conducted additional large-scale experiments on **ImageNet-C**, comparing our method against popular TTA baselines including **TENT, ETA, MEMO and ActMAD**. The results are **[PRESENTED IN THIS PLOT](https://imgur.com/a/imagenet-corruptions-robustness-bGaru0u)**, which will be included in the final version of the paper.
>
> As shown in the plot, our method consistently outperforms the baselines across all severity levels and batch sizes, achieving a 3–5% improvement in accuracy. These results further highlight the effectiveness and scalability of our approach.
>
> &nbsp;
>
> ### **"For segmentation results, could the authors include additional evaluations on commonly used benchmarks such as KITTI-C and nuScenes? This would enable a fairer comparison, as the dataset used in this paper has not been widely adopted in previous works.**"
>
> Thank you for pointing out these benchmarks. They are relevant. We missed them because prior TTT works don't use them. We intend to remedy this when revising but, unfortunately, training on them would take more time than we have for this rebuttal. Thus, to provide additional results during the rebuttal period, we prioritized experiments on ImageNet-C, as discussed in our response to the previous comment.
>
> &nbsp;
>
> ### **"The error bars in Figures 3, 4, and 8 are difficult to recognize and distinguish, particularly for color vision-deficient readers."**
>
> In the revised version, we will improve the clarity and accessibility of the error bars, following the approach used in **[PLOT](https://imgur.com/a/imagenet-corruptions-robustness-bGaru0u)**. Specifically, we made the error bars bolder and more visible, used brighter, more distinguishable colors, and added explicit labels to differentiate the bars. These improvements will be applied to all relevant figures in the paper, along with better-written captions to clearly explain the contents of each plot and guide interpretation—ensuring improved readability, including for color vision-deficient readers.
>
> &nbsp;
>
> ### **"Additionally, could the authors provide more qualitative results instead of bar charts? The absolute differences in bar heights are not intuitive to interpret."**
>
> When revising, we will add an Appendix. D with some more qualitative segmentation results for the aerial photography road segmentation challenge. You can view these qualitative results via the **[FOLLOWING THIS ANONYMOUS LINK](https://imgur.com/a/JCOWYgq)**.
>
> In the figure, we display the segmentation predictions of TTT, ActMAD, and IT$^3$ on OOD data. Quality scores are added on top of each prediction because they offer an objective measure of segmentation performance, helping to interpret the results more intuitively.
>
> &nbsp;
>
> ### **"Could the authors provide a computational complexity analysis comparing their method with existing baselines?"**
>
> Runtime analysis can be found in Appendix A. When revising, we will add analysis of memory consumption to all cases as well.
> One case for batch size 128 ImageNet-C, peak memory reserved with torch.cuda.max_memory_reserved() in GB:
>
> TENT: 4.8
> ETA: 4.9
> MEMO: 13.5  (3 augmentations)
> ActMAD: 7.2
> IT$^3$: 7.4
> Vanilla Model: 4.5

---

> > ### Comment · Reviewer_4J9m · 2025-04-08
> >
> > Thanks for the authors' response. My main concerns have been addressed, and I would like to keep my original score.

---

### Official Review · Reviewer_WBsB · 2025-03-13

**Overall Recommendation:** 3

**Summary:**

The authors:
1. Make the claim that enfocing idempotence is benificial for test-time training tasks.
2. Design a paradigm that brings an auxiliary signal representing ground truth, and force the network to learn idempotence through minimizing $\Vert f_{\theta}(x, y) - y \Vert + \Vert f_{\theta}(x, 0) - y \Vert$.
3. Conduct experment on various dataset and shows promising results.

**Claims And Evidence:**

All of the claims are generally adequate.

**Essential References Not Discussed:**

N/A

**Experimental Designs Or Analyses:**

The experiment covers various aspect of tasks, including image recognition, tabular prediction, which is overall convincing promising.
One issue, however, is line 235 in Experiments section: "we include the original TTT method and a newer more versatile approach". It’s unclear which auxiliary task "the original TTT" refers to, and whether other auxiliary tasks were evaluated. I hope the authors can clarify this point.

**Methods And Evaluation Criteria:**

The evaluation benchmark is comprehensive and provides convincing evidence to show the effectiveness of proposed method.

**Other Comments Or Suggestions:**

Section 2.1: The argue "TTT operates per-instance, with no assumption that future test data will be similar. So previous work have treated TTA and TTT as distinct paradigms" seems weak. As some works, e.g. [1] and [2] do consider TTT under online setting.

[1] Gandelsman, Y., Sun, Y., Chen, X., and Efros, A. Test-time training with masked autoencoders.

[2] Renhao Wang, Yu Sun, Yossi Gandelsman, Xinlei Chen, Alexei A Efros, and Xiaolong Wang. Test-time training on video streams.

**Other Strengths And Weaknesses:**

Strenths:

1. The idea of bring idempotence to test-time training is innovative.
2. The experiments is comprehensive.

Weaknesses:
1. The paper's presentation lacks clarity. Specifically, one of its key claim is "if such a network is trained so that f(x, 0) = f(x, y) = y, then at test time the distance ||f(x, f(x, 0))−f(x, 0)|| correlates strongly with the prediction error". However, the authors only skim through it, which disrupts the logical flow. Although efforts are made on explaning how to keep idempotence during pretraining and test-time training phase, they should provide further explanation and intuition on why enforcing idempotence is beneficial for enhancing performance.
2. Similarly to point (1), I'm interested to see ablation studies examining the impact of enforcing idempotence within the TTT framework.

**Questions For Authors:**

The authors should reorganize the presentation, as detailed in my "Other Strengths And Weaknesses" section. For the remaining points, the strengths and weaknesses are clear and straightforward in my view, so I probably won't change my score.

**Relation To Broader Scientific Literature:**

This paper discusses and brings a now paradigm for test-time training. Which might be instructive for future work and paradigm design in this field.

**Theoretical Claims:**

I checked the derivation and discussion of idempotence in main paper and appendix, and they are adequate.

---

> ### Author Rebuttal · Authors · 2025-04-01
>
> We sincerely thank the reviewer for their thoughtful and constructive feedback. Below we address all the comments, and will revise the camera-ready version accordingly.
>
> &nbsp;
>
> ### **"The paper's presentation lacks clarity. Specifically, one of its key claim is "if such a network is trained so that f(x, 0) = f(x, y) = y, then at test time the distance ||f(x, f(x, 0))−f(x, 0)|| correlates strongly with the prediction error". However, the authors only skim through it, which disrupts the logical flow."**
>
> Thank you for identifying this important issue. We acknowledge that the paper lacks a clear logical flow connecting idempotence to OOD adaptation. We provide here **[A SKETCHED LOGICAL CHAIN](https://imgur.com/XwHPhbc)** explaining exactly this. The introduction will be rewritten accordingly and a detailed extended version will be added to the paper as appendix C. In short:
>
> 1. When using ZigZag approach to training, the discrepancy between recursive calls of the model, $|| y_1 - y_0 ||$, is highly correlated to how far out-of-distribution (OOD) an input is.
> 2. Thus, we can use $|| y_1 - y_0 ||$ as a loss, and reduce 'OODness'.
> 3. By doing this, we make the network more idempotent: $L_{\mathrm{IT^3}} = || y_1 - y_0 || = || f(x, f(x,0)) - f(x,0) ||.$
> 4.  But properly minimizing $L_{\mathrm{IT^3}}$ is non trivial, so we employ the approach from IGN (Shocher et al.)
> 5. This can be understood as a projection onto a distribution. (Appendix B).
>
> &nbsp;
>
> ### **"I'm interested to see ablation studies examining the impact of enforcing idempotence within the TTT framework."**
>
> We provide empirical evidence analyzing the impact of enforcing idempotence in Fig. 2 of the main paper. It shows histograms of the distance from idempotence across four scenarios. For the original training and validation data, the model demonstrates very high idempotence (with the validation set slightly lower than the training set). For Out-of-Distribution (OOD) data, the model initially shows near-random idempotence. However, after applying our TTT optimization, the distribution of OOD data is significantly "shifted" from this random state towards that of the training and validation sets. This supports our claim that optimizing for idempotence indeed makes the model behave on OOD data similarly to how it behaves on in-distribution data.
>
> To further investigate this, we conducted additional experiments on ImageNet-C, measuring idempotence loss across various corruption types and severity levels. **[(SEE ANALYSIS FIGURE HERE)](https://imgur.com/a/idempotence-vs-imagenet-performance-ym0McKq)**. Each point in the plot represents a batch of data; the x-axis shows the computed idempotence loss for that batch, while the y-axis indicates its classification accuracy. The data covers 15 corruption types and 5 severity levels, which are visualized using the colorbar. As expected, higher severity levels of corruption correspond to larger idempotence loss, further validating the effectiveness of enforcing idempotence within the TTT framework and highlighting its strong negative correlation with model performance (Pearson correlation: –0.94).
>
> &nbsp;
>
> ### **"Section 2.1: The argue "TTT operates per-instance, with no assumption that future test data will be similar. So previous work have treated TTA and TTT as distinct paradigms" seems weak. As some works, e.g. [1] and [2] do consider TTT under online setting."**
>
> We conducted a new large-scale experiment on ImageNet-C that also includes comparisons to TTA methods TENT and ETA across various batch sizes. **([SEE RESULTS HERE](https://imgur.com/a/imagenet-corruptions-robustness-bGaru0u))** These will be added to the main paper in the camera-ready version.
>
> As the reviewer points out, there is previous work [1, 2] about the online setting that we present in sections 3.3 and 4.6. However, unlike in [1,2], ours is intended for continual learning. Furthermore, we want to make a distinction between TTA, along with its many variants, and online-TTT: While the latter is used  in a data-streaming context the former is used on a given dataset or batch, or the training data at hand.
>
> &nbsp;
>
> ### **"In line 235 in the Experiments section: It's unclear which auxiliary task 'the original TTT method' refers to or whether other auxiliary tasks were evaluated."**
>
> Our apologies for this lack of clarity. By original TTT, we meant Sun et al. 2020. The auxiliary task used in their work is orientation prediction for images, with validation on image recognition benchmarks. We also compare to ActMAD (Mirza et al., 2023), which we adapted for other types of data and tasks.
>
> &nbsp;
>
> ### **"The authors should reorganize the presentation, as detailed in my "Other Strengths And Weaknesses" section."**
> Thank you for helping us improve the presentation of our paper. We will reorganize accordingly.

---

> > ### Comment · Reviewer_WBsB · 2025-04-04
> >
> > Thanks to the authors for addressing my concerns. As previously mentioned, I would like to maintain my score.

---

### Decision · Program_Chairs · 2025-05-01

**Decision:**

Accept (poster)

**Comment:**

The authors propose a method for test-time training. The main claim is in the assumption of idempotence as the primary mechanism for updating the model parameters at test time and builds on existing work [A, B] and extends it towards test-time training.

While the reviewers came to a positive consensus, we note that the paper bares similarity to [C], where both utilizes the outcome from an uninformative input as an auxiliary supervision signal and enforces consistency as the primary mechanism for adaptation. However, [C] was not mentioned.

We also note that reviewer tgc2 had doubts on whether it made sense that the approach should improve OOD robustness and noted that the results are largely empirical, tgc2 ultimately improved their score despite the concern.

[A] Shocher et al.  Idempotent generative network. ICLR 2024.

[B] Durasov et a. Zigzag: universal sampling-free uncertainty estimation through two-step inference. Transactions on Machine Learning Research, 2024.

[C] Park et al. Test-time adaptation for depth completion. CVPR 2024.